# Efficiency Assessment of an Amended Oscillating Water Column Using OpenFOAM

**Mobin Masoomi** [1]**, Mahdi Yousefifard** [1] and **Amir Mosavi** [2,3,4,]*

1   Department of Mechanical Engineering, Babol Noshirvani University of Technology, Babol 47184-71167, Iran;
    m.mo.masoomi@gmail.com (M.M.); yousefifard@nit.ac.ir (M.Y.)
2   Faculty of Civil Engineering, Technische Universität Dresden, 01069 Dresden, Germany
3   John von Neumann Faculty of Informatics, Obuda University, 1034 Budapest, Hungary
4   Information Systems, University of Siegen, 57072 Siegen, Germany
*   Correspondence: amir.mosavi@mailbox.tu-dresden.de

**Abstract:** Oscillating water column (OWC) is an advanced form of wave energy converter (WEC). This study aims at improving the efficiency of an amended OWC through a novel methodology for simulating several vertical plates within the chamber. This paper provides a numerical investigation considering one, two, three, and four vertical plates. The open field operation and manipulation (OpenFOAM) solver is verified based on the Reynolds-Averaged Navier–Stokes (RANS) equation. Results show the number and the position of plates where the convertor's efficiency improves. The work undertaken here also revealed a reduction in the net force imposed on the convertor's structure, especially the front wall. Consequently, adding plates acquires more efficiency with lower force on the system.

**Keywords:** oscillating water column; vertical plates; hydrodynamic; structural forces; OpenFOAM; computational fluid dynamics; mechanics; simulation; computation; structural mechanics

## 1. Introduction

The oscillating water column (OWC) is an advanced energy converter consisting of two essential parts. The first part is the main chamber, and the second is the power take-off system [1]. These two functional and well-integrated systems are coupled to generate energy efficiently. Apart from the power take-off system, the main chamber characteristics involve the chamber width discussed in the present study [2]. On the one hand, the device efficiency depends on incident waves; thus, the related change in the device geometry could increase the efficiency. On the other hand, the alterations' schemes should be applicable besides considering the additional costs to perform the alterations [3]. The OWC's performance is greatly influenced by free surface motion and internal air pressure; thus, accurate and low-cost analysis of the energy converters was performed under different conditions due to improving the numerical methods. Reliable computational fluid dynamics (CFD) simulations can capture detailed flow physics, especially near the chamber and free surface [4]. Although some researchers have investigated the multi-chambers [1] and L-shaped duct [2] to improve the efficiency by changing the chambers' width and sequential, in the present paper, the innovative idea is using the fixed vertical rectangular plates inside only one chamber, which decreases the free surface's sloshing effects and increases the device's efficiency over a particular wave number.

Many researchers have evaluated changes in the inner chambers, sea bed, or surrounding walls. The first fundamental research in this field was presented by Maeda et al. [3], who introduced a simple prediction method for absorbing wave power characteristics. Takahashi et al. [4] also tested a full-scale model converter. They focused on resisting stability improvement and performance as a breakwater by absorbing the wave energy. Howe and Nader [5] investigated the influence of change in inlet geometries of two kinds

of bent OWCs; they concluded that the inlet geometry variations provided low deviations in the output power. Numerical analysis of a nearshore OWC by considering the role of stepped bottom topography was presented by Rezanejad et al. [6]; their study showed that the added step at the sea bottom might significantly increase the device's efficiency and power absorption. Ashlin et al. [7] investigated an array of OWC devices by considering the different distances' effect on hydrodynamic performance; their results proved that an array of OWCs has better efficiency than a single and isolated device. He and Huang [8] studied pile-supported breakwaters and the effects of OWC characteristics (width, free surface water level, an orifice) on OWC energy extraction and free surface fluctuation inside the chamber. Elhanafi et al. [9] experimentally and numerically measured the wave force on a 3D offshore stationary OWC. They calculated forces on an OWC under different wave conditions. Xu and Huang [10] presented a 3D CFD simulation of circular OWC with nonlinear power take-off. They examined the resonant sloshing inside the pneumatic chamber.

Ning et al. [11] compared the results of the single- and dual-chamber OWC and found that the efficiency of the dual chambers remains more stable at bigger frequency intervals than the standard single-chamber model. Elhanafi et al. [12] analyzed the performance of solitary and dual-chamber OWC numerically. They found that compared with single-chamber OWC, it could increase efficiency by about 40% over intermediate and long wave periods. Recently, Simonetti et al. [13] presented an empirical model to optimize a stationary OWC device; this model can be used in the initial step of OWC design. Teixiera et al. [14] conducted a numerical simulation to analyze the turbine's OWC device and aerodynamic model's performance. Pawitan et al. [15] used OWC as a traditional vertical breakwater: They presented loading distribution measurements for a large-scale OWC. Lauro et al. [16] studied a stability analysis of a wave energy converter using experiments and numerical solutions. Moreover, they studied the pressure distribution and loads due to these pressures on the converter. Luo et al. [17] investigated the 2D nonlinear solution using numerical wave tanks to calculate the efficiency of fixed OWC. Ashlin et al. [18] studied the extent to which the bottom profile shape affected the OWC device's efficiency. They found that the OWC efficiency decreased with the high steepness wave due to the bottom profile inside the chamber and reflected water from the lip wall. Connell et al. [19] developed a dynamic OWC with nonlinear PTO interaction; they used a numerical wave tank to simulate 1-DOF spar buoy OWC. Raj et al. [20] experimentally researched the influence of the opening angle of walls and effective resonant length; the relative capture width (RCW) can be 75%, more than the OWC without harbor walls. Rezanejad et al. [21] performed an OWC device analysis in the stepped bottom condition. They implemented OpenFOAM (open-source field operation and manipulation) code based on fully nonlinear Reynolds-averaged Navier–Stokes (RANS) equations. They also perused the water particle pattern near the step. The loads imposed on the fixed offshore OWC were studied by Elhanafi [22] using the CFD approach. Kamath et al. [23] used the porous media theory to model the top opening of the OWC: They found that the efficiency curve at low wave steepness remains stable. Morris-Thomas et al. [24] experimentally evaluated the OWC device to calculate the energy absorption and total efficiency for power take-off. The natural frequency range for the best-recorded efficiency was a value near 70%, based on their study.

The OpenFOAM solvers are used widely in wave-structure interactions, Devolder et al. [25] used an IHFOAM extension to simulate wave run-up around a monopile encounter with incident waves. Hu et al. [26] used the new wave boundary condition at OpenFOAM to model the interaction of incident waves with a fixed/floating truncated cylinder and simplified floating production storage and off-loading platform (FPSO). Hu et al. [27] investigated how OpenFOAM performs for wave interactions with a vertical cylinder by a complete comparison against experimental results for three regular waves. Some research was conducted using OpenFOAM at the scope of OWCs' wave-structure interactions. Iturrioz et al. [28] presented one of the first studies in this field that used this software; they performed a numerical simulation on the hydrodynamic behavior of an

OWC and validated their results with experimental data. Deng et al. [29] studied the fixed offshore structure with a flat horizontal plate at the bottom using a wave2Foam extension for wave generating. Simonetti et al. [30] proposed a new dimensionless parameter for a fixed OWC device to provide generalized equations delivering correction factors for OWCs. The current study aims to offer a simple and effective way to improve the wave energy converter's performance at the design point.

In the present study, InterFoam, based on the Volume Of Fluid method (VOF), was used to simulate the effects of different wave conditions in a numerical wave tank (NWT) to evaluate the hydrodynamic characteristics of the amended conventional OWC. That, vertical free surface movements, internal chamber pressure, OWC hydrodynamic efficiency, and forces acting on the device were computed and compared against experimental measurements. The preference of OpenFOAM is both due to its open-source licensing and high accuracy for the case in hand. The amended converter's performance was evaluated under various conditions and compared to the original OWC.

## 2. Material and Methods

### 2.1. Governing Equations

In the present study, the RANS equations used to capture the turbulent flow characters' by considering the Reynolds stresses besides satisfying the continuity and mass conservation equation. The turbulence model is used as an auxiliary equation in a time-average model [31], the fluid is considered an incompressible viscous fluid with constant and stable properties, and the formulation is represented in the Einstein summation convention approach.

$$\frac{\partial u_i}{\partial t} + \frac{\partial}{\partial x_j}\left(u_i u_j\right) = -\frac{\partial p}{\partial x_i} + v\frac{\partial^2 u_i}{\partial x_j \partial x_i} \tag{1}$$

$$\frac{\partial u_i}{\partial x_i} = 0 \tag{2}$$

The symbol $t$ represents time and $x_i$, $x_j$ are $x$, $y$, $z$ in short. Furthermore, $U_i$, $p$, and $v$ are the fluid velocity, pressure, and fluid kinematic viscosity, respectively. A key factor to updating the Navier-Stokes (N-S) into the RANS equation is using the Reynolds decomposition, which involves the decomposition of the two dependent variables (velocity and pressure) within two parts, mean and fluctuating value, as represented in Equation (3):

$$u_i = \overline{u_i} + u_i' p = \overline{p} + p' \tag{3}$$

The flow substantially involved two generic variables, $\phi$, and $\psi$, and the mean value for each of them can be calculated by considering the time average $(\overline{\phi}^T)$ or volume average $(\overline{\phi}^V)$ for steady-state and spatially homogeneous flow, respectively, in a turbulent regime [31]. By using the dominant relations for the mean mathematical operators, the new generation for $u_i$ and $p$, extracted from Equation (3), which are substituted into Equations (1) and (2). The differential equation is obtained based on the mean pressure and velocity for the turbulent and incompressible flow regime:

$$\frac{\partial \overline{u_i}}{\partial t} + \frac{\partial}{\partial x_j}\left(\overline{u_i u_j}\right) = -\frac{\partial \overline{p}}{\partial x_i} + v\frac{\partial^2 \overline{u_i}}{\partial x_j \partial x_i} \tag{4}$$

$$\frac{\partial \overline{u_i}}{\partial x_i} = 0 \tag{5}$$

A further substitution for the nonlinear term $\overline{u_i u_j}$ and using this convective term as the nonconservative form $\tau_{ij}$. The governing equations, named RANS equations, generated and used as the main Equation of the numerical solution [32]:

$$\overline{u_i u_j} = \overline{\left(\overline{u_i} + u_i'\right)\left(\overline{u_j} + u_j'\right)} = \overline{u_i}\,\overline{u_j} + \overline{\overline{u_i}u_j'} + \overline{u_i'\overline{u_j}} + \overline{u_i'u_j'} = \overline{u_i}\,\overline{u_j} + \overline{u_i'u_j'} \tag{6}$$

$$\frac{\partial \overline{u_i}}{\partial t} + \overline{u_j} \frac{\partial \overline{u_i}}{\partial x_j} = -\frac{\partial \overline{p}}{\partial x_i} + \nu \frac{\partial^2 \overline{u_i}}{\partial x_j \partial x_i} - \frac{\partial \tau_{ij}}{\partial x_j} \tag{7}$$

$$\frac{\partial \overline{u_i}}{\partial x_i} = 0 \tag{8}$$

The Reynolds-stress tensor incorporates the motions, and the mean value for stresses in the framework of a symmetric matrix involving normal stresses as the diagonal array and shear stresses placed at off-diagonal components. The interface between the two fluids requires special formulation to maintain a sharp interface; numerical diffusion would otherwise mix two fluids over the whole domain [33]. Another significant parameter assigned is the choice between implicit and explicit schemes; this is wholly discussed in Versteeg and Malalasekera [34]. Based on the governing equations, the pressure and velocity are coupled with each other, that, the pressure is solved and used to guess and improve the velocity; then, the improved velocity is used to calculate the pressure. This loop is continued until convergence and the correct answer is reached.

The Reynolds-stress tensor adds six additional independent unknowns and expresses as a function of mean variables; thus, the turbulent models are engaged to decrease the order of velocity fluctuations. There were three different approaches for the turbulent regime: one-, two-, and three-equation models. The two-equations model is commonly used due to its simple implementation for RANS equations [35,36]. In particular, the $k - \varepsilon$ model which is a two-equation model consider for the present study formulation. The turbulent effect is described with the viscosity variations on the flow regime. By considering the Reynolds stresses represented as:

$$\tau_{ij} = -\rho \overline{u_i' u_j'} = \nu_t \left( \frac{\partial \overline{U_i}}{\partial x_j} + \frac{\partial \overline{U_j}}{\partial x_i} \right) - \frac{2}{3} k \rho \delta_{ij} \tag{9}$$

$$\nu_t = c_\mu \frac{k^2}{\varepsilon} \tag{10}$$

where $\nu_t$ is the turbulent viscosity, $C_\mu$ is a constant parameter, $k$ is the turbulent kinetic energy, and $\varepsilon$ is a dissipation rate for $k$; a further discussion is mentioned in Launder and Spalding [37]. By substituting the turbulence equations into RANS equations, the partial differential equation changes into the new form, as represented in Equations (11) and (12).

$$\frac{\partial \varepsilon}{\partial t} + \overline{u_i} \frac{\partial \varepsilon}{\partial x_i} = -c_{\varepsilon 1} \frac{\varepsilon}{k} \tau_{ij} \frac{\partial \overline{u_i}}{\partial x_j} + \frac{\partial}{\partial x_i} \left( \frac{\nu_t}{\sigma_\varepsilon} \frac{\partial \varepsilon}{\partial x_i} \right) - c_{\varepsilon 2} \frac{\varepsilon^2}{k} + \nu \frac{\partial^2 \varepsilon}{\partial x_i \partial x_i} \tag{11}$$

$$\frac{\partial k}{\partial t} + \overline{u_i} \frac{\partial k}{\partial x_i} = -\tau_{ij} \frac{\partial \overline{u_i}}{\partial x_j} - \varepsilon + \frac{\partial}{\partial x_i} \left( \frac{\nu_t}{\sigma_k} \frac{\partial k}{\partial x_i} \right) + \nu \frac{\partial^2 k}{\partial x_i \partial x_i} \tag{12}$$

A complete system of partial differential equations involving Equations (7), (8), (11), and (12) is a closed system with some constant values. These values are extracted from a comparison with physical experiments; the values of the constants are:

$$c_{\varepsilon 1} = 1.44, \ c_{\varepsilon 2} = 1.92, \ c_\mu = 0.09, \ \sigma_k = 1.0, \ \sigma_\varepsilon = 1.3 \tag{13}$$

Since the standard $k - \varepsilon$ model can not be integrated with the cases with a solid boundary, the $c_\mu$ is considered as a constant parameter to establish the consistency of the results with the wall condition. The methods are fully explained in Patel et al. [38], and Launder et al. [39].

### 2.2. Model Geometry and Boundary Conditions

The OWC's numerical domain is a simple 2D rectangle with an inlet and outlet to generate waves. The overall wave tank length is five times the wavelength with a constant height of 2 m. The distance between the converter and the wave inlet is sufficiently large to

eliminate the reflected-wave effects. As mentioned earlier, in this study, a 2D numerical model scale of 1:12.5 is considered, with initial dimensions rather than the validated case by Kamath et al. [23], shown in Table 1. The cross-section of the original OWC device is illustrated schematically in Figure 1. The top, side, and bottom of the domain are adopted with total pressure inlet/outlet, symmetry, and no-slip boundary conditions. The OWC's body is set to the wall with a no-slip boundary condition. The incident air and water inlet velocity are set to zero.

**Table 1.** Geometrical details of the standard OWC.

| Main Dimensions | Symbol | Model | Real | Unit |
| --- | --- | --- | --- | --- |
| OWC height | H | 1.275 | 15.93 | [m] |
| Chamber breadth | B | 0.64 | 8 | [m] |
| OWC thickness | C | 0.05 | 0.625 | [m] |
| Chamber lip height | a | 0.15 | 1.875 | [m] |
| Water depth | d | 0.92 | 11.5 | [m] |
| Orifice width | dB | 0.005 | - | [m] |

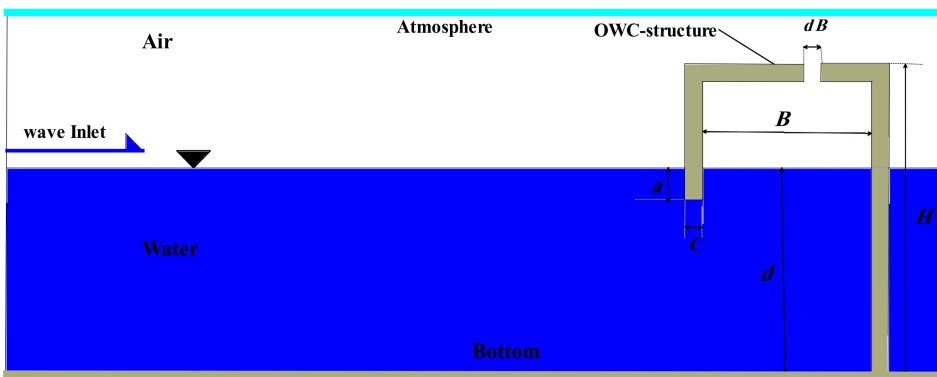

**Figure 1.** The standard OWC layout and dimensions symbols.

### 2.3. Wave Generation Scheme

The methods related to wave generation at OpenFOAM are divided into two main groups. The first is using extensions such as IHFOAM [40–42], OlaFoam [43,44], or Wave2Foam [45,46], which are added to standard libraries. The second one uses the standard solvers by considering the physical flap or in-house code correlate with the boundary conditions [47,48]. For the present study, a standard model is used with the newly developed sets, by which new boundary conditions are added for wave generation and absorption with a low-cost and simple method. Reliable wave generation and stability can be achieved by an additional feature added to the InterFoam solver from OpenFOAM version 5.0 onwards. Wave parameters such as wave frequency, wave amplitude, and the appropriate domain depth are needed to set the wave generation scheme at the WaveDict folder.

The fifth-order wave equation is more complicated than the Airy wave so that the wave's harmony can be better recognized for numerical studies. Albeit, apart from the seawater level (SWL) (downward), the behavior of fifth-order waves is similar to the linear wave theory [49]. The fifth-order theory involves five members in a series form in which each member is smaller than the previous one. One of the important primary parameters is the horizontal velocity calculated with Equation (14) with a five-term series. The analysis is performed under different wavelengths to estimate the device efficiency for a wide range of working conditions. Therefore, a dimensionless parameter, "Kd," is used as a symbol for evaluating wave characteristics, as formerly used in Morris-Thomas et al. [24] and Kamath et al. [23].

$$u = \sum_{n=1}^{5} u_n . \cosh(nks) . \cos n(kx - \omega t) \tag{14}$$

$$k = \frac{\omega^2}{g} \tag{15}$$

where $t$ stands for the wave propagation time, $\omega$ is the angular frequency, k is the angular wave-number, $g$ is the gravity, and finally, s = a + d, in which d is the still water depth and a is the wave amplitude. A complete version of the dominant equations is referenced in Chakrabarti [50]. Apart from the free surface level and the region near the structure, fifth-order behavior is more similar to a linear wave. The incident wave characteristics are identified in Table 2.

**Table 2.** Propagation wave characteristics.

| Kd [–] | Unit | 0.52 | 0.7 | 1.26 | 1.8 | 2.5 | 3.5 |
|---|---|---|---|---|---|---|---|
| Wave length | [m] | 7.36 | 6.1 | 4.07 | 3.0 | 2.29 | 1.5 |
| Wave height | [m] | 0.12 | 0.12 | 0.12 | 0.12 | 0.12 | 0.12 |
| Wave period | [s] | 2.73 | 2.3 | 1.71 | 1.43 | 1.21 | 1.0 |

*2.4. Mesh Convergence Analysis*

At the first step, the mesh generation process is shaped based on two pre-processing tools for OpenFOAM, BlockMesh, and snappyHexMesh. The procedure in the present study is such that BlockMesh generated the based grid size for the whole domain, and after that, some refinement zones are added to the base mesh using the snappyHexMesh tool. As shown in Figure 2, the cells near the free surface are refined to have an accurate wave generation. These refinements are also applied for the cells inside and outside the OWC, especially around the orifice due to its small dimension. The advantage of using this strategy is generating structured mesh with high quality and a low number of cells.

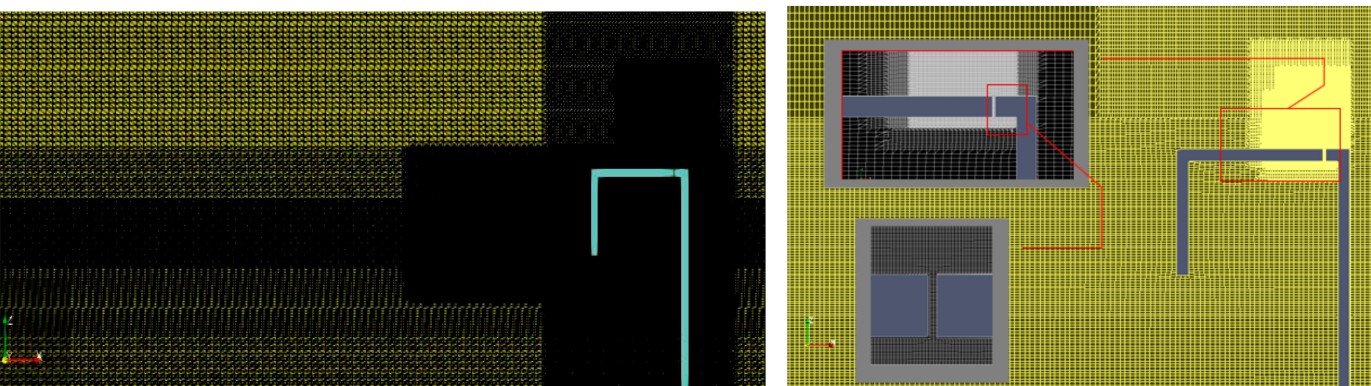

**Figure 2.** The structured mesh around the OWC body and orifice.

The main framework of the griding strategy is to ensure the accuracy of the generated wave in the numerical wave tank, besides preventing wave damping. The considered wave, wave amplitude = 0.06 m and a wavelength = 4.07 m (Kd = 1.26 $\xi$ = 0.029); and the free surface depth is considered to be d = 0.92 m. The numerical tank has three different grid sizes (*dx* = 0.1 m, 0.05 m, 0.025 m), as shown in Figure 3. Free surface-level oscillation is captured in different positions in the numerical domain, x = 5, 10, 15, and 17 m, from the inlet. Since the initial dimension of griding cells is small, no significant improvement occurred in wave generation. The lines for the theory in Figure 3 come from the wave amplitude, and the free surface height from the bottom, lower bond = 0.92 − 0.06 = 0.86 m, and upper bond = 0.92 + 0.06 = 0.98 m are specified to evaluate the accuracy of the numerical solution. Although each case's results are similar, to get accuracy with the low computational cost, dx = 0.05 m is a reasonable choice for the case in hand.

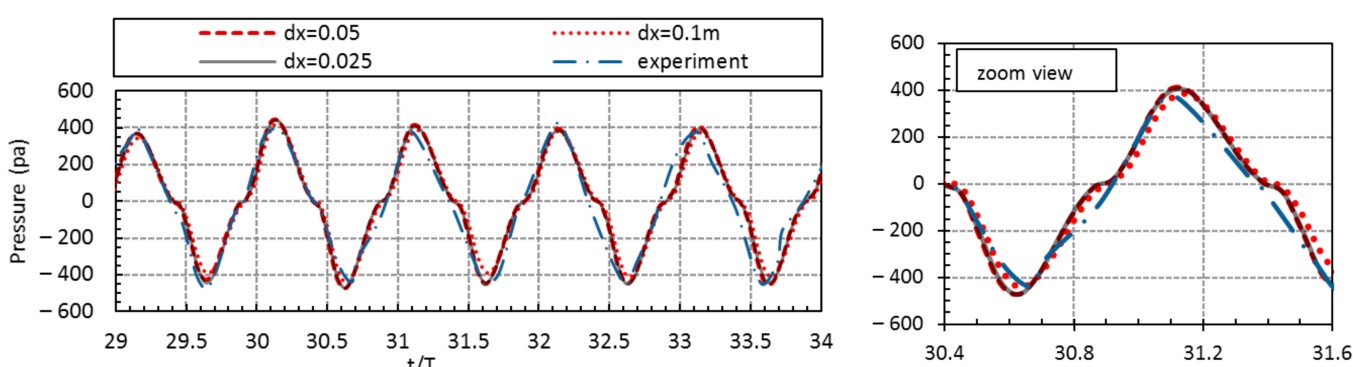

**Figure 3.** Mesh study considers wave amplitude at four different positions (x = 5, 10, 15, 17 m).

Another variable used to check the accuracy of the meshing strategies is the pressure discrepancies inside the chamber; pressure oscillations for Kd = 1.26 are recorded at a point, x = 18.7 m, z = 1.12 m via time, as illustrated in Figure 4. As can be inferred, no marked difference existed between different grid sizes, and the experimental results came from Kamath et al. [23]; thus, dx = 0.05 m is an appropriate grid size satisfy the accuracy needed and low cost of simulation.

**Figure 4.** Grid convergence study by considering the pressure for Kd = 1.26.

### 2.5. Validation of the Numerical Model

The determinant parameter for most investigations performed on wave energy converters is evaluating the hydrodynamic efficiency, which comprises the relation between the inlet wave energy and the outlet energy flux based on the following Equation:

$$P_{in} = \frac{\rho_w \cdot g \cdot h^2 \cdot \lambda}{16T} \left[ 1 + \frac{4\pi d/\lambda}{\sinh(4\pi d/\lambda)} \right] \qquad (16)$$

where $h$ and $\lambda$ are the wave height and length, and d is the water depth within the OWC chamber. Equation (16) indicates the theoretical wave-induced power for the OWC. The power available at the vicinity of the orifice, $P_{out}$, is measured in the following Equation:

$$P_{out} = \frac{1}{T} \int_0^T P_c(t) \cdot q(t) dt = \frac{1}{2} |P_c| \cdot |wfs| b \cdot l \cos(\theta) \qquad (17)$$

where $\theta$, $wfs$, and $P_c$ are the phase difference, free surface velocity, and air pressure, respectively. Finally, the hydrodynamic efficiency is governed by the inlet power ratio (from incident waves) and the orifice's output power.

$$\eta = \frac{P_{out}}{P_{in}} \qquad (18)$$

The validity of the two-phase and free-surface VOF algorithm in OpenFOAM is evaluated rather than the represented results by Kamath et al. [23] and Morris-Thomas et al. [24], for similar incident wave character, 4.07 m length and 1.71 wave period. Figures 5 and 6 compare pneumatic pressure and vertical surface velocity component results for Kd = 1.26 and Kd = 0.52, respectively. Since the incident wave's manner is periodic, the coincidence of the resulted pressure and free surface velocity is a bit different for each wave period. Thus, an average error needed to be represented as a cumulative error.

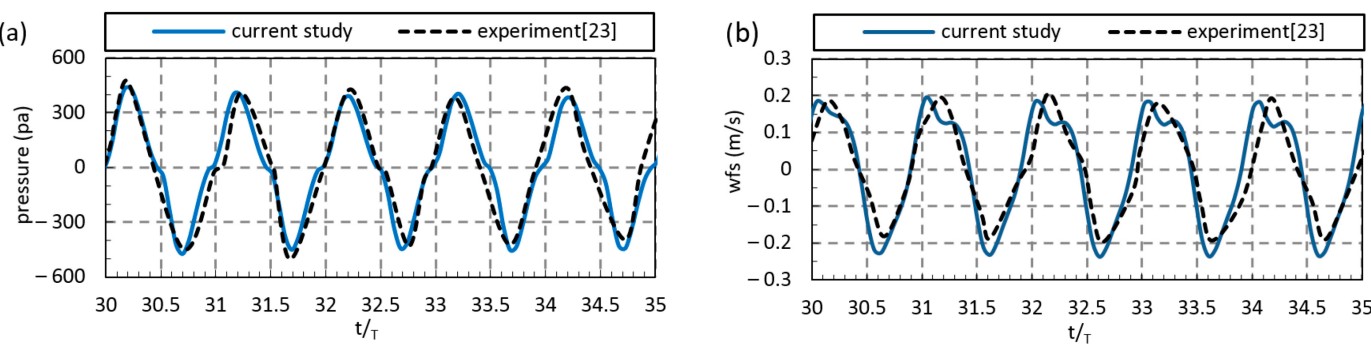

**Figure 5.** The variation of pneumatic pressure inside the chamber (**a**) and vertical component of free surface velocity (**b**) versus non-dimensional time (Kd = 1.26).

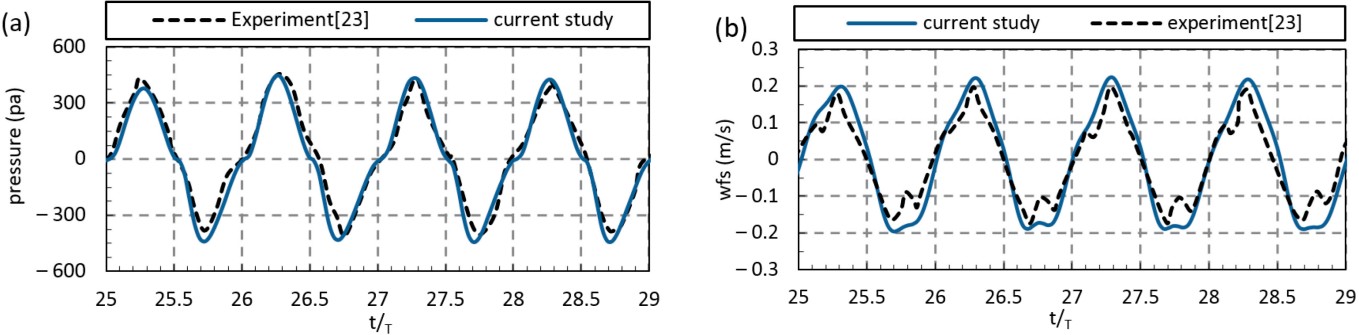

**Figure 6.** The variation of pneumatic pressure inside the chamber (**a**) and vertical component of free surface velocity (**b**) versus non-dimensional time (Kd = 0.52).

Each converter has its own natural frequency related to the built geometry and dimensions. Furthermore, the device's hydrodynamic efficiency is highly dependent on the incident wave's parameter. Thus, the efficiency analysis needed to be performed for different wavelengths, presented in Figure 7. The maximum converter efficiency is achieved at Kd = 1.26, the design point of the system. The total converter efficiency discrepancies between the present and experimental results are about E = 6.5% on average.

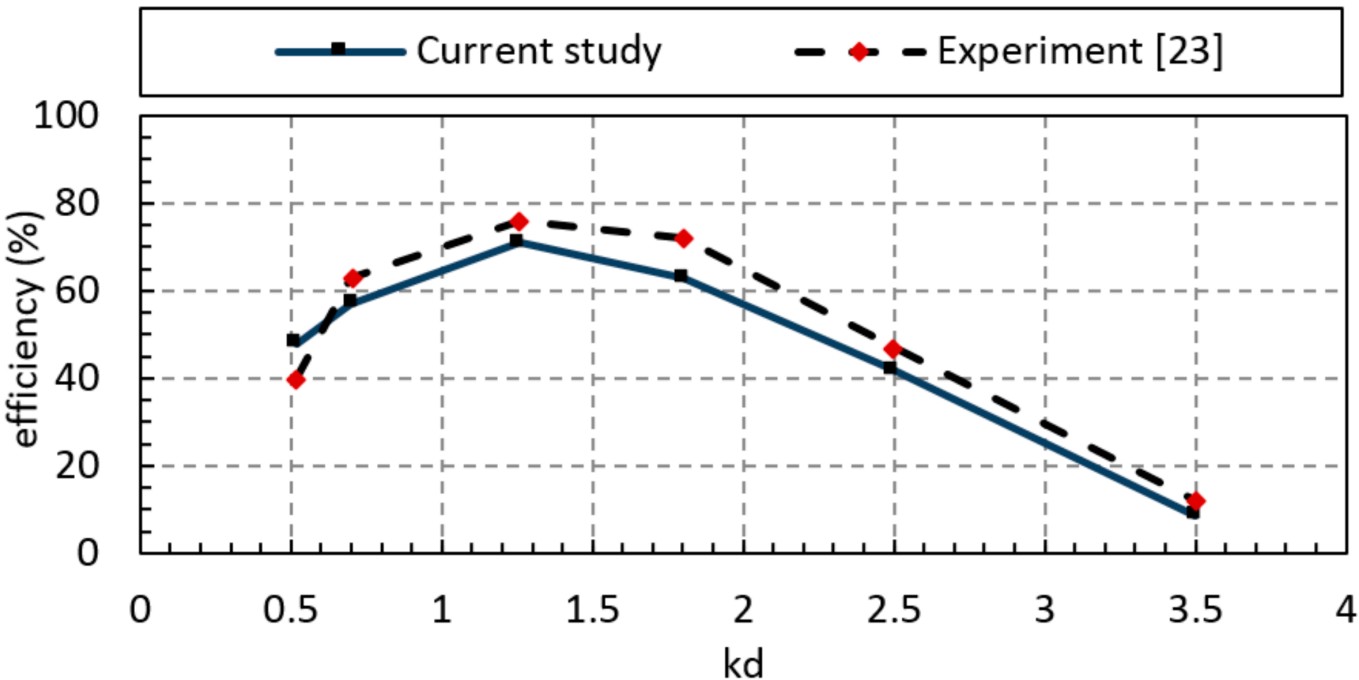

**Figure 7.** Comparison of numerical and experimental efficiencies for different wavelengths (Kd).

From Kd = 0.52 to Kd = 1.26, the device efficiency increased until it reaches a maximum value at the system design point; from Kd = 1.26 to Kd = 3.5, the efficiency reduced significantly. These operational ranges are also highly dependent on the values for B/d and B/a, b: chamber breath, d: chamber depth, a: front wall height. In other words, the motion inside the chamber could be represented as a rigid piston for lower B/d or B/a, and whatever these ratios increased, the piston-like motion disrupted due to a change in the resonance frequency of the system, as mentioned in Evans and Porter's [51].

In the current study, for B/d = 0.7, the OWC devices' resonance occurred at Kd = 1.26, which is discussed further in the following sections. A wave period is divided into eight equal parts, and the air velocity contours are presented for different time steps in Figure 8. However, the sloshing phenomenon's effects on the range of free surface oscillations are negligible; irregular free-surface fluctuations caused a deviation from its piston-like motion.

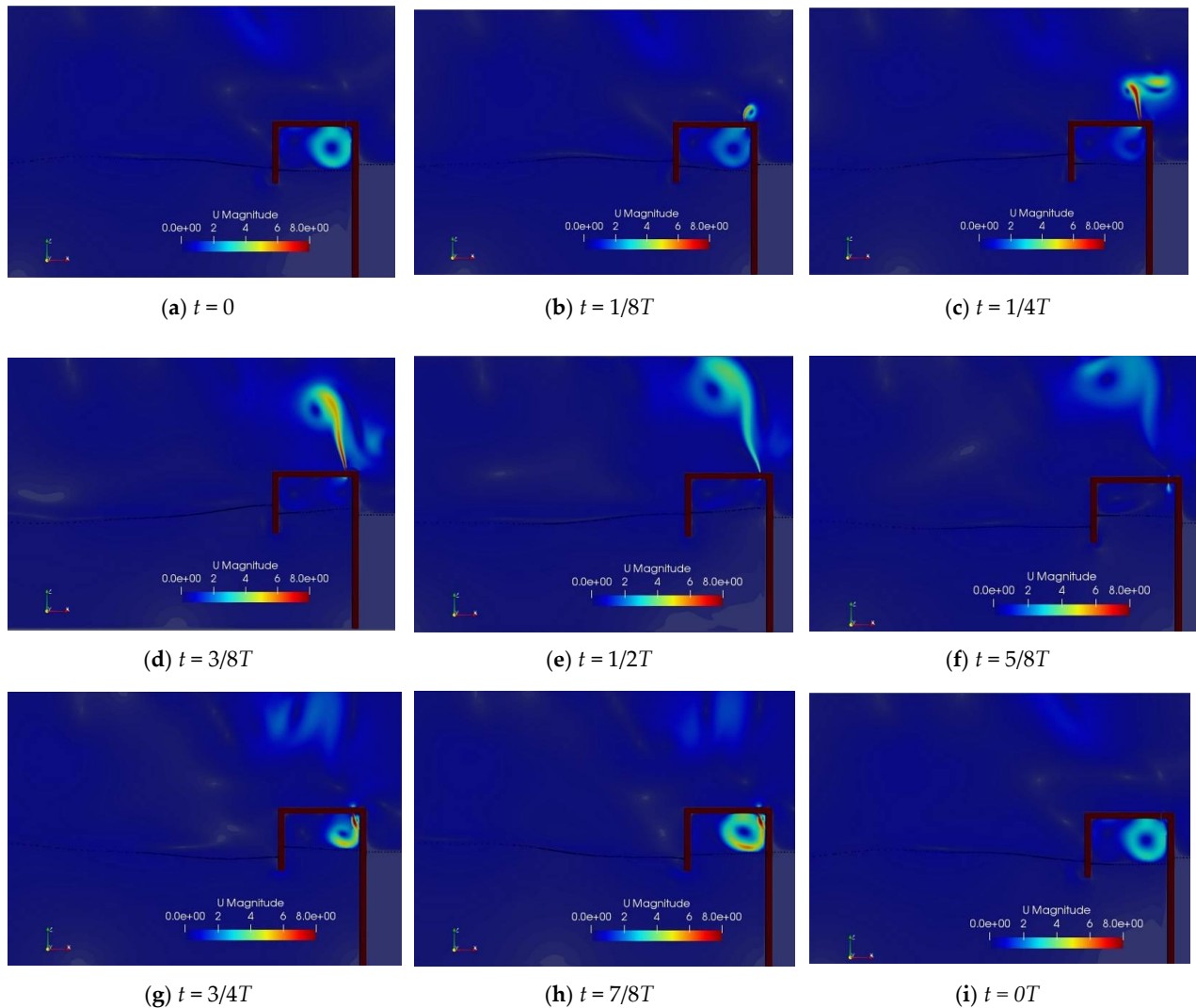

**Figure 8.** Air velocity contour for different time steps at a particular wavelength (Kd = 1.26).

## 3. Results and Discussion

Multi-chamber OWCs are mainly involved sub-chambers in an array. For instance, O'Boyle et al. [52] experimentally investigated the wave-field variations around arrays of OWCs; they inferred that the wave-particle distribution is relative to the arrangement of OWCs and the proportion of wavelength to multi-chamber spacing. The arrays of OWCs studied using the FEM method to investigate the variation of hydrodynamic efficiency were proposed by Nader et al. [53]. They found that the neighboring chambers in an array caused particular effects on each other and significantly influenced the total efficiency. Even for large separations, the output air from the orifice for OWCs in an array may have different pressure and velocity value rather than a solitary OWC with the same dimensions.

Rezanejad et al. [54] found that a dual-chamber OWC device on the stepped sea bottom can improve efficiency in a wide range of wave amplitudes compared with the single-chamber case. Wang et al. [55] investigated the hydrodynamic efficiency of the dual-chamber OWC device numerically. The hydrodynamic efficiency is calculated by both the air pressure and free surface elevation inside the chamber. Two sub-chambers in an array with a particular scheme can extract more power than a solitary chamber. In fact, the entrance water from generated waves moves distinctly in each sub-chamber that affects the resonant frequency [56,57].

### 3.1. Amended OWC by Considering Added Vertical Plates

To investigate a comprehensive case, one, two, three, and four vertical plates are added into the interior part of a standard OWC along the chamber width. The chamber is divided into an equal number of sub-chambers, illustrated in Table 3, to enhance the performance of the OWC device in this paper. As shown in Figure 9, some vertical plates are added to the inner part of the chamber with only one orifice, without any change in the total width of the OWC. In fact, the resonance frequency is highly dependent on the water mass inside the chamber; the plates' thickness is considered small enough to have an unchangeable water mass within the chamber. Therefore, the resonance frequency is expected to remain constant. The second issue is the free surface movements; two types exist, the sloshing and the piston mode, the lowest and highest efficiencies are recorded for sloshing mode and piston mode, respectively. To better illustrate this phenomenon, three vertical plates with four equal sub-chambers are shown in Figure 9; this figure shows the upgrade device's status with the gauges' location to extract hydrodynamic parameters.

**Table 3.** Specification of added plates case study.

| Case Study | Symbol | I | II | III | IIII | Unit |
|---|---|---|---|---|---|---|
| Number of plates | - | 1 | 2 | 3 | 4 | [–] |
| Height of plate | z | 0.3 | 0.3 | 0.3 | 0.3 | [m] |
| thickness of plate | dx | 0.02 | 0.2 | 0.02 | 0.2 | [m] |
| Sub-chamber width | b | 0.31 | 0.2 | 0.145 | 0.112 | [m] |

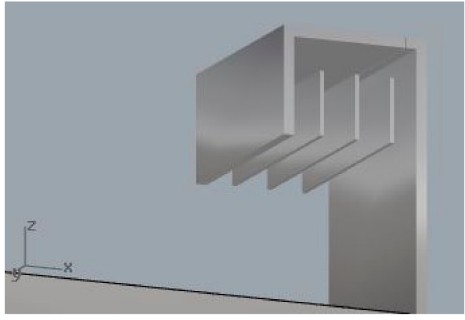 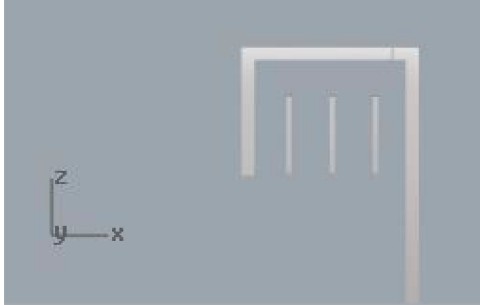

**Figure 9.** A sample schematic view of amended OWC (three plates/OWC).

However, the free surface motion and the related velocity are captured from the middle of the chamber [24,58]; the question is whether this assumption is appropriate for the multi-chamber devices or not? The integral and center-point methods are two different schemes for calculating volume of water in each sub-chamber and recording the free surface [59]. Although the relative difference between the two methods is not significant and regularly less than 5%, the center-point method is used in the present study:

$$\frac{(EL_n \cdot b_n) - V_n}{V_n} \text{ (number of the chamber)} \tag{19}$$

where $n$ is the chamber number, $EL_n$ is the free surface elevation, and $b_n$, $V_n$ are the width and volume of each sub-chamber, respectively. As mentioned earlier, the vertical plate installation is done to divide the chamber's interior volume into equal parts. Different wave characteristics, shown in Table 2, are used to study the effects of adding plates on the maximum free surface elevation for the G-sensors, Figure 10. Adding plates to the standard OWC caused an increment in the vertical free-surface elevation values for all Kds', except Kd = 0.52. The most significant increment for free surface velocity is observed at Kd = 1.26, about 20%. The evaluations are illustrated in Figure 11 for the standard model and three plates/OWC.

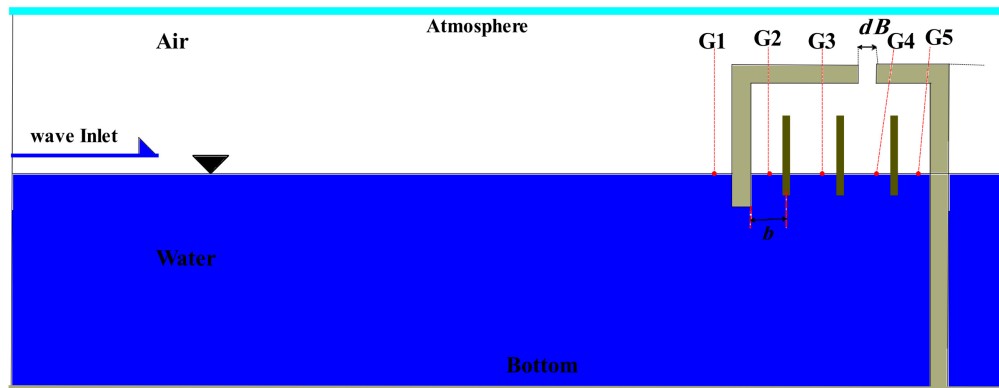

**Figure 10.** A sample of amended OWC and the position of wave gauges inside the chamber.

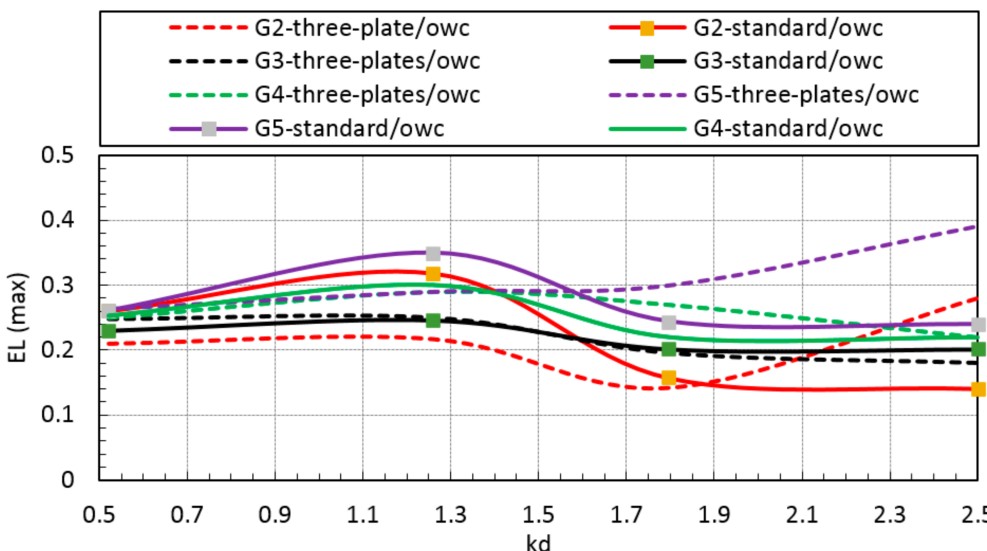

**Figure 11.** The maximum free surface elevation, EL [m] for each sensor (G1–G5) at different wavelengths (Kd).

To better illustrate the effect of the added plate, a contour base comparison between standard and amended OWC performed, shown in Figure 12. The velocity dispersion for the inner and outer parts of the chamber changed. For instance, the dispersion near the orifice for standard and one-plate OWC is similar, and the velocity concentration is near the orifice. However, for the others, the velocity dispersed inside the chamber, and there is no significant concentration near the orifice. In addition, for the outer part of the OWC, the dispersion of minimum and maximum concentrations are different.

*3.2. Total Efficiency Assessment*

The efficiency is more akin to three parameters; pressure, free surface velocity, and phase difference, which can be affected by adding vertical plates. Therefore, further analysis is needed to assess how or to what extent the three parameters' variations caused more or fewer converters' efficiency. The most efficient case study should; increase output pressure, increase free surface velocity, and decrease the phase difference between pressure and free surface velocity. In the following sections, these parameters are discussed in more detail.

### 3.2.1. Free Surface Velocity Evaluation

There are two primary sources for wave mobility in the sub-chambers: transmitted and reflected waves. By adding the vertical plates within the chamber, the portion of reflected waves increased. Free surface velocity oscillations are presented at three locations; the first point is near the front wall (x = 18.45 m), the second point is at the center (x = 18.7 m), and the third point is adjacent to the end wall of the OWC (x = 18.9 m). As shown in Figure 13, the free surface velocity value (wfs) for four plates/OWC is more than the others. For a point near the back wall (x = 18.9 m), the reflected wave effect is not noticeable; thus, adding plates does not affect free surface velocity discernibly. Therefore, the standard OWC experienced higher values.

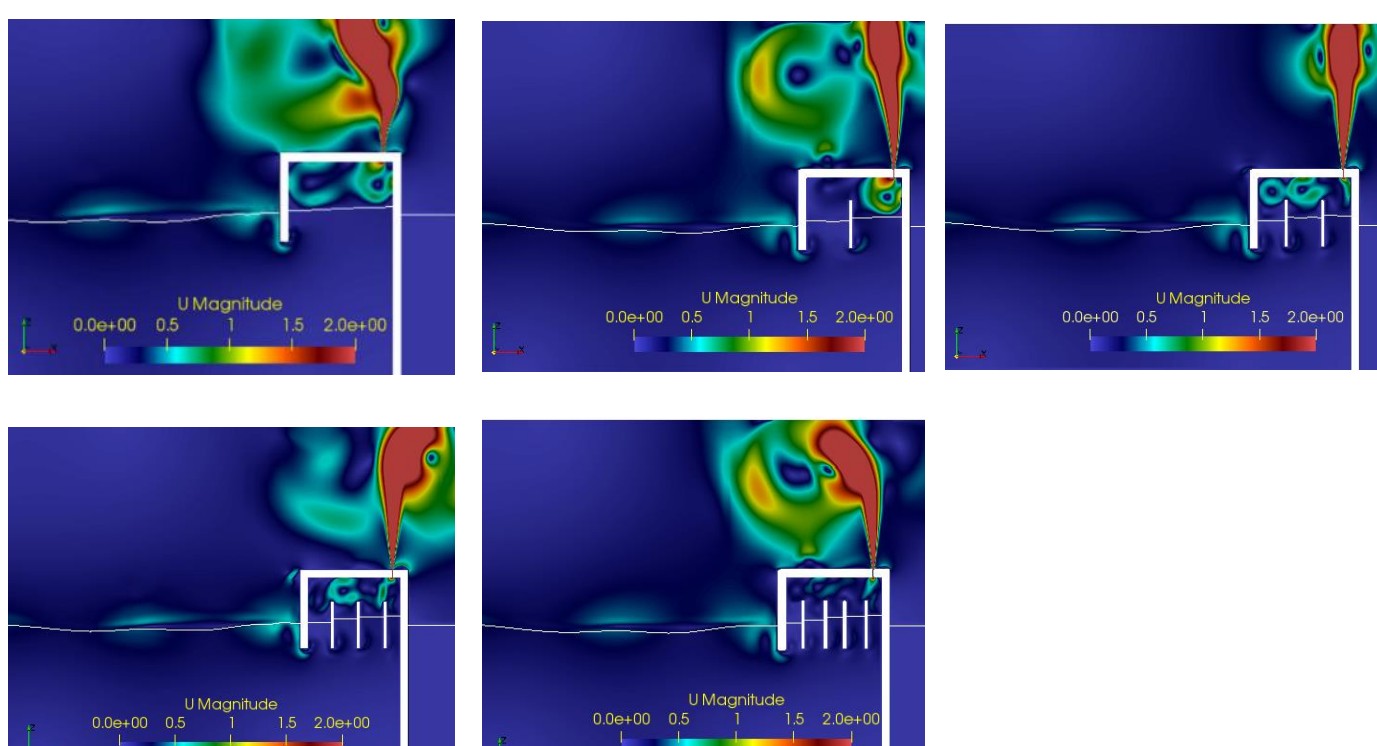

**Figure 12.** Velocity contour for Kd = 18 at T = 15.6 s (exhalation step) for different number of internal vertical plates.

As can be inferred from Figure 13, the two peaks/bottoms phenomenon occurred for standard OWC for all three sensor locations and one plate at the x = 18.45 location. The reason is the width of the free surface; in fact, the manner of the wider free surface is not as smooth as the manner for narrower ones. The free surface in each sub-chamber became smoother by adding vertical plates, with lower quasi-sloshing, which caused smoother peaks and bottoms for its diagrams.

This phenomenon changed the inhalation/exhalation phase, which causes the more extensive phase difference between pressure and free surface velocity. Although Figure 13 only represents the diagrams for Kd = 1.26, the same goes for the rest of Kd's such that; whatever the wavelength became smaller, Kd > 1.26, the free surface became more turbulent; thus, this phenomenon became more discernible. Accordingly, it can be inferred that adding vertical plates could alleviate the harshness of the free surface to modify the peaks and bottom of the diagram, which directly correlate with the phase difference and efficiency, which is discussed further in the following sections.

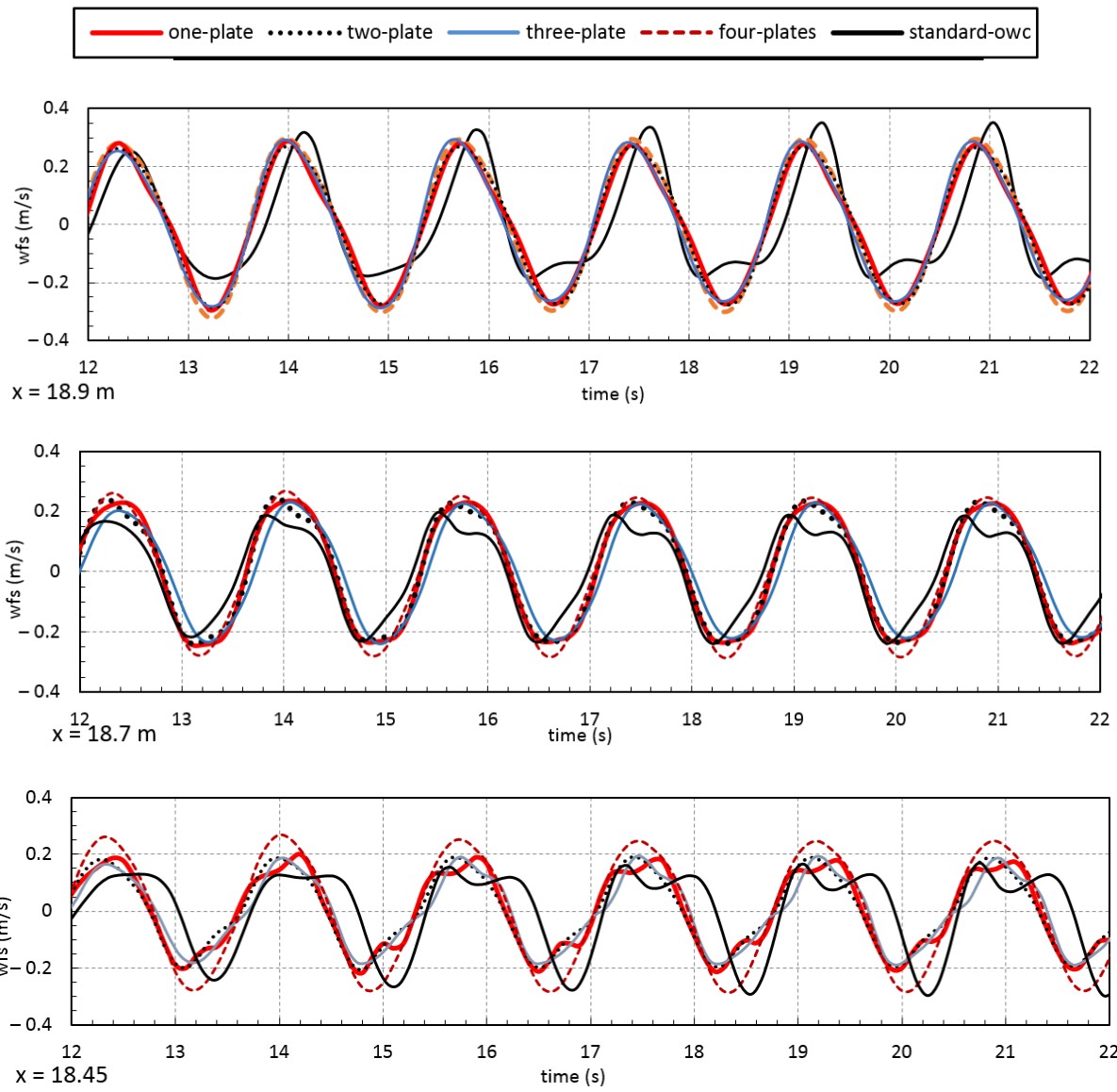

**Figure 13.** Free surface velocity for the standard model and multi-plate model at different positions in the chamber (Kd = 1.26).

### 3.2.2. Output Pressure Evaluation

The effects of adding plates on the maximum pressure inside the chamber are presented in Figure 14. As can be inferred, the maximum pneumatic pressure increment inside the chamber for all cases is about 5.8% for Kd = 1.26. The pressure discrepancies for different wavelengths do not indicate uniform behavior. Although for the two wavelengths Kd = 0.7 and Kd = 1.6, the maximum pressure values increased, the pressure decreased for other wavelengths, the valid question is why the pressure does not change significantly? The answer is "the chamber's volume and the free surface elevation." Since the vertical plates are considered thin enough, the chambers' volume remained almost constant; thus, the amount of air at the inhalation and exhalation phase does not change significantly, causing stability for the output pressures from the orifice apart from the number of vertical plates. No more power is extracted based on pressure increment; therefore, the pressure variations do not significantly change efficiency.

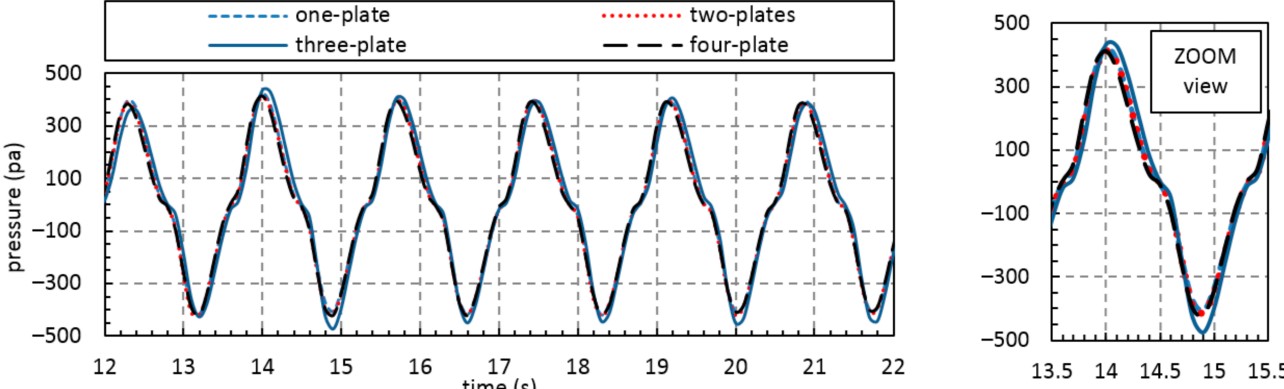

**Figure 14.** Pneumatic pressure changes for amended OWC (Kd = 1.26).

### 3.2.3. Phase Difference Evaluation

One of the most critical parameters is the phase difference between pressure and free surface velocity. The water particle excursions are the determining factor, whereas at Kd = 1.26, with a smaller wavelength than Kd = 0.52, the water particles have a smaller excursion in the chamber [51]; thus, the free surface motions are smoother and more uniform within the chamber. This physical phenomenon helped the multi-chamber OWC; as illustrated in Figure 15, a case with three plates had a smaller excursion, especially at T = 19.6 s, which resulted in a smoother free surface and subsequently lower values for the phase difference.

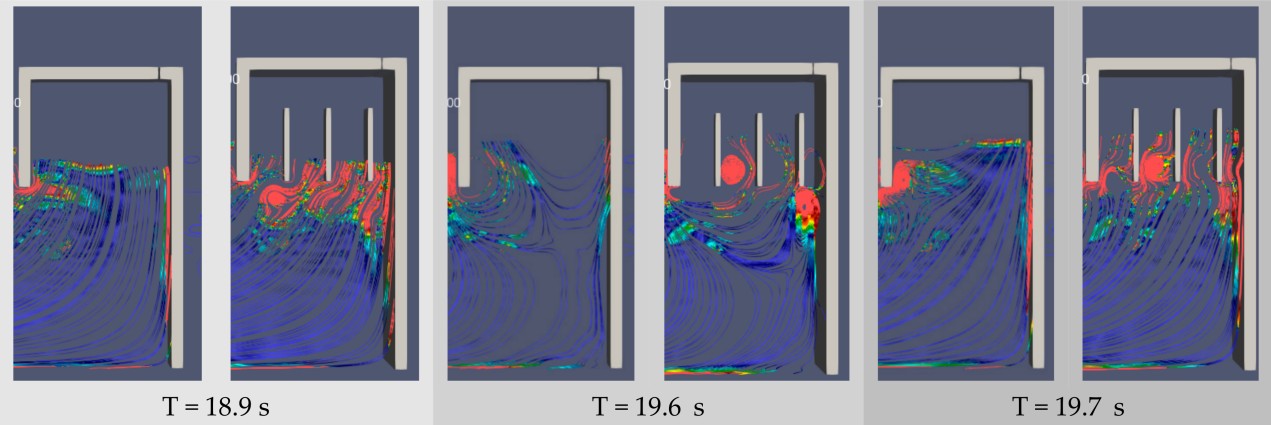

**Figure 15.** Water particle excursion for the standard model and three-plate OWC at Kd = 1.26.

The underlying physics to calculate the phase difference is based on the difference between two consecutive peaks along the time axis $T_a$-$T_b$, then the calculated time differences $T_s$ are divided into one period, T. The obtained number β is considered to evaluate the phase difference θ, as illustrated in Equation (20) step by step from (a) to (c). All case studies had their special phase differences, but only three-plate OWC is represented due to the paper's brevity. As shown in Figure 16, the phase difference for three plates/OWC decreased; this trend is true for other cases.

$$T_a - T_b = T_s \tag{20a}$$

$$\frac{T}{T_S} = \beta \tag{20b}$$

$$\frac{360}{\beta} = \theta \tag{20c}$$

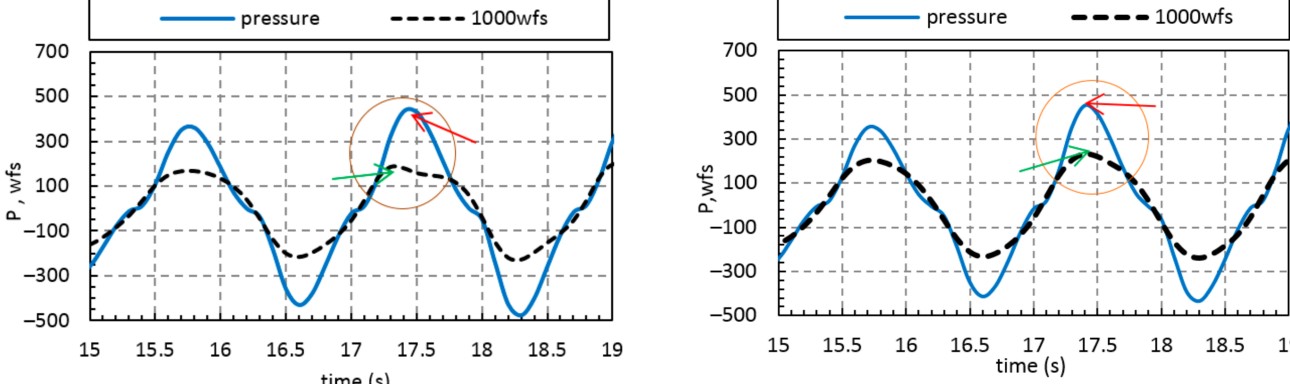

**Figure 16.** The pressure oscillation inside the converter [Pa] and the free surface velocity [m/s × 1000] related to standard OWC (**left**) and three-plate OWC (**right**) (Kd = 1.26).

As previously said in Equation (17), the output power is directly related to the value of $\cos\theta$, in fact, A lower phase difference results in more power extraction. For the case illustrated in Figure 16, the standard OWCs phase difference is $\theta = 45°$, which reduced to $\theta = 25°$ For the amended OWC, Kd = 1.26. The physical inference is the change in the inhalation and exhalation phase, which is more compatible with the free surface motion and piston-like motions. The phase difference values for all amended OWCs, with one, two, three, and four plates, are about $\theta = 20° - 25°$ On average, at Kd = 1.26, which is the most efficient wavelength for this convertor.

### 3.2.4. Total Efficiency Evaluation

The converter efficiency increased by a maximum value of 17% by considering the represented three parameters, especially in the convertor's resonant range, Kd = 1.26. At the system's design point, the converter's total efficiency increased from e = 71% to e = 88% after adding the vertical plates. Adding more than three plates did not significantly affect the efficiency, as illustrated for each case in Figure 17.

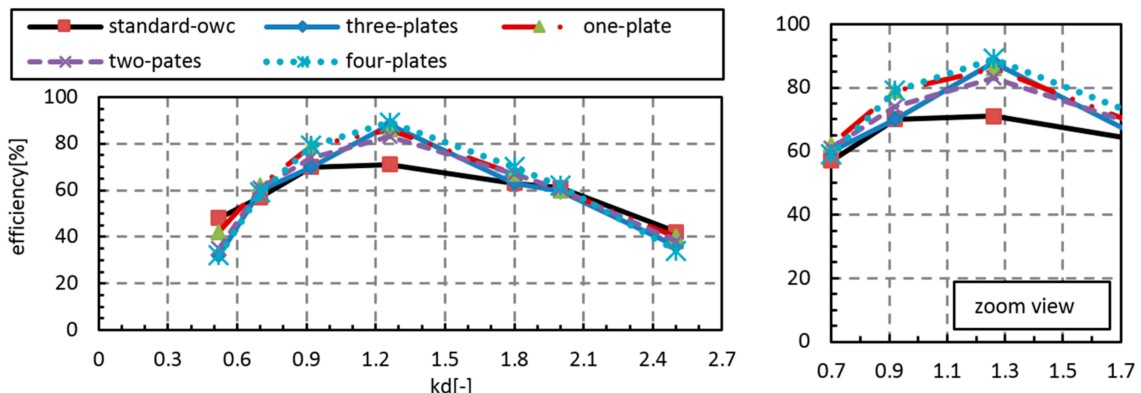

**Figure 17.** Total efficiency recorded from the amended OWC at different wavelengths and different Kd.

An effective mechanism whereby all parameters can be evaluated is using a comprehensive symbol. For this reason, the disparity symbol is used to illustrate each parameter's deviation of amended OWC rather than standard OWC. The disparity percentage is calculated by substituting the three primary parameters, p: pressure, wfs: free surface velocity, and E: efficiency for standard and amended OWC in Equation (21). This approach is performed for each case study, with one, two, three, and four plates/OWC gathered as diagrams in Figure 18. For three plates/OWC, particularly from Kd = 0.7 to Kd = 1.5, the pressure increased but has a reverse trend for other cases. Apart from the number of plates, the efficiency increased from Kd = 0.7 to Kd = 1.8. That is to say, the hydrodynamic

performance of the device can be improved for some specific wave frequencies that are near the natural frequency.

$$\frac{p_{\text{amended}} - p_{\text{standard}}}{p_{\text{standard}}} = \text{disparity\%} \tag{21a}$$

$$\frac{\text{wfs}_{\text{amended}} - \text{wfs}_{\text{standard}}}{\text{wfs}_{\text{standard}}} = \text{disparity\%} \tag{21b}$$

$$\frac{e_{\text{amended}} - e_{\text{standard}}}{e_{\text{standard}}} = \text{disparity\%} \tag{21c}$$

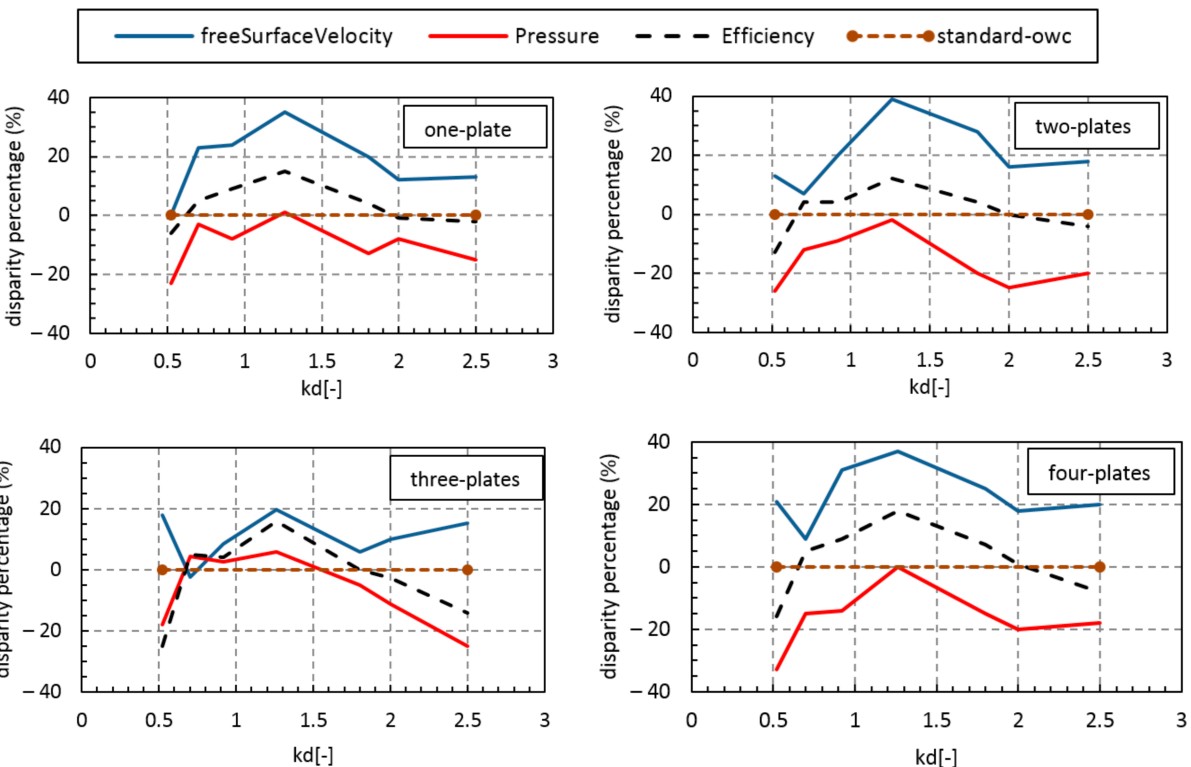

**Figure 18.** Disparity comparison between the standard model and amended OWC for different wavelengths and different Kd.

### 3.3. Structural Force

A review of this field research confirmed that the main focus is to improve the hydrodynamic efficiency, and less attention has been paid to the structural loads on an OWC. Wave-induced stress can have devastating effects on specific parts of the structure of the converter. For example, the front and top walls of an OWC chamber are more critical than other sections. So far, limited research has been done on calculating the force applied to the structure of an OWC. Jayakumar [60] examined the force applied by the wave on the converter using an experimental method. Ashlin et al. [18] investigated how the incident waves induced force on the fixed-shore OWC; two main parameters are involved in their experiments, wave steepness and inside depth of the OWC.

Didier et al. [61] evaluated the wave height influence on the force imposed on the OWC's structure, particularly the front wall, by using the smoothed particle hydrodynamics (SPH) method. Ashlin et al. [62] investigated the dynamic pressure and the structural force experimentally: They concluded that the vertical forces are smaller than the horizontal forces. Viviano et al. [63] carried out several experiments on forces acting on the front wall of OWC devices. Ning et al. [64] performed the experimental and numerical analysis of wave-induced forces on OWC structure; they concluded that the exerted forces are highly

dependent on wave characteristics. The waves with larger heights increase the force; in contrast, longer waves decrease the force imposed on the structure. The most similar investigation rather than the case in hand was Ning et al. [65], which was experimentally investigated the force on a dual-chamber OWC besides the hydrodynamic characteristics.

Recently, a study that assessed the influence of viscosity on the wave-induced force was performed by Wang and Ning [66]. A viable solution to recognize the OWC structure's force is integrating the pressures with normal surface vectors. There are two choices in OpenFOAM to reach an accurate result for structural force. First, generating the surface area and surface normals of the OWC and multiplying with the pressure using ParaView, which is the graphical and post-processing interface of OpenFoam. By considering this method, the real force on the OWC can be calculated rapidly without solving the numerical solution again. The calculated force is the cumulative value of the hydrostatic and hydrodynamic loads caused by the air, water, and pneumatic pressure inside the chamber. Second, the calculation of force can be implemented using a force function (functionObjectLibs ("libforces.so") added to controlDict, and then the force and pressure are calculated simultaneously at each time step. Forces comprising normal pressure $F_p$ and tangential viscous contributions $F_v$ are shown in Equation (22).

$$\begin{aligned} F_p &= \sum \sigma_i \cdot S_{f,i}(p_i - p_{ref}) \\ F_v &= \sum s_{f,i}(\mu \cdot R_{dev}) \end{aligned} \tag{22}$$

where $\sigma_i$ is the density, $S_{f,i}$ the face area vector, $p$ is the pressure, $\mu$ is the dynamic viscosity, and $R_{dev}$ is the deviatoric stress tensor. The accuracy and value for the pressure are verified in the previous sections, and extra verification related to wave force is not required. Figure 19 shows the force variation on the device versus time for different Kd. As can be seen, the wavelength variations significantly affected the force acting on the OWC, as discussed below. As expected, longer waves caused a more prominent force to the OWC's structure.

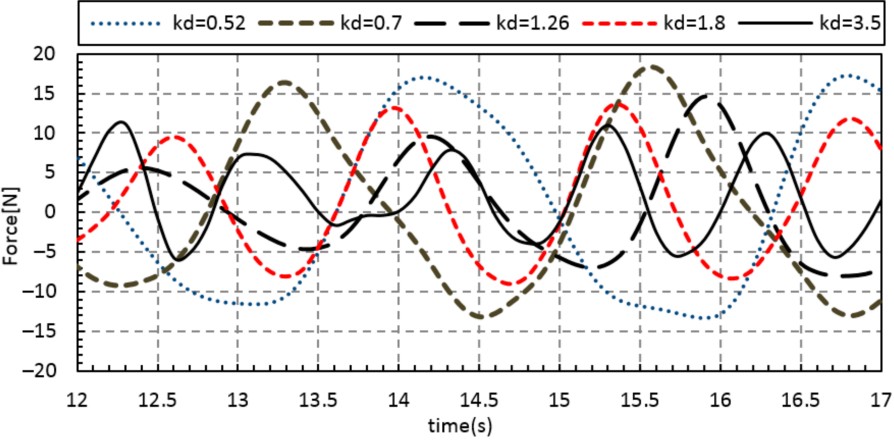

**Figure 19.** Horizontal force values acting on a standard OWC for different Kd.

The forces applied on the OWC in both cases (standard and multi-plates/chamber) are compared, the overall force applied to the system is the summation of the forces applied to the body and plates, $F_{total} = F_{body} + F_{plates}$. Figure 20a demonstrates the hull/structure and the plate's forces from Kd = 0.52 to Kd = 2.5; the hull's forces slightly decreased at the system's design point, Kd = 1.26, and the forces acting on the OWC with three plates are shown in Figure 20b. the consideration is that, although the total amount of force applied to the system does not change significantly, the force applied to each converter element substantially changed, such that installing vertical plates inside the chamber could reduce the portion of force applied to the convertor's hull to a large extent.

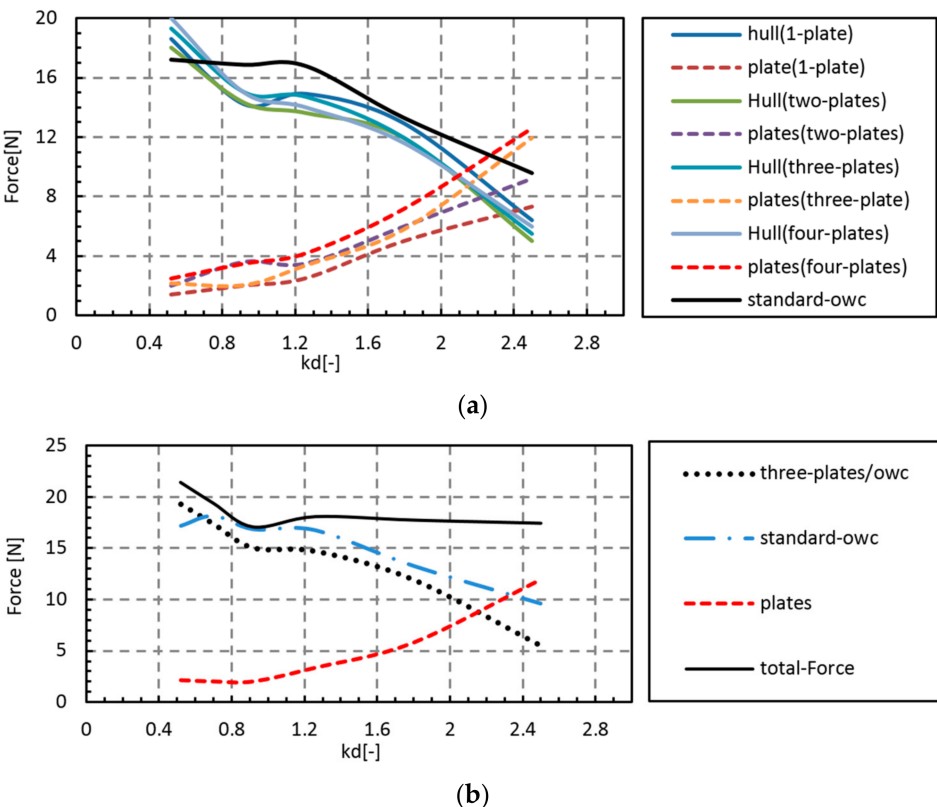

(a)

(b)

**Figure 20.** Horizontal force on the multi-plates/OWC (**a**) and three plates/OWC (**b**) at different Kd.

As illustrated in Figure 21, depicting force oscillations for two plates/OWC at Kd = 1.26; the most significant portion of the OWC structure's force belonged to incident waves encountered on the front wall and main structure, and more than half of the force applied to the converter is exerted on the front wall. The valid question is, to what extent does add vertical plates affect the wave-load inducement on the front wall? It is evident from Figure 22, adding vertical plates caused lower values for wave loads on the front wall. By increasing the plates, less force is applied to the front wall, such that, among these considered cases, almost a 30% decrease occurred by adding four plates inside the chamber versus the standard model. Therefore, it can be concluded that the OWC with vertical plates able to reduce the forces applied to the OWC and improve the structural strength.

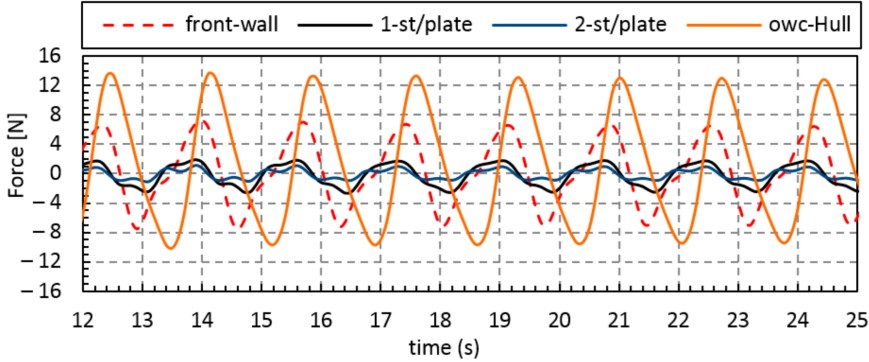

**Figure 21.** Comparison of the force applied to two plates/OWC for Kd = 1.26.

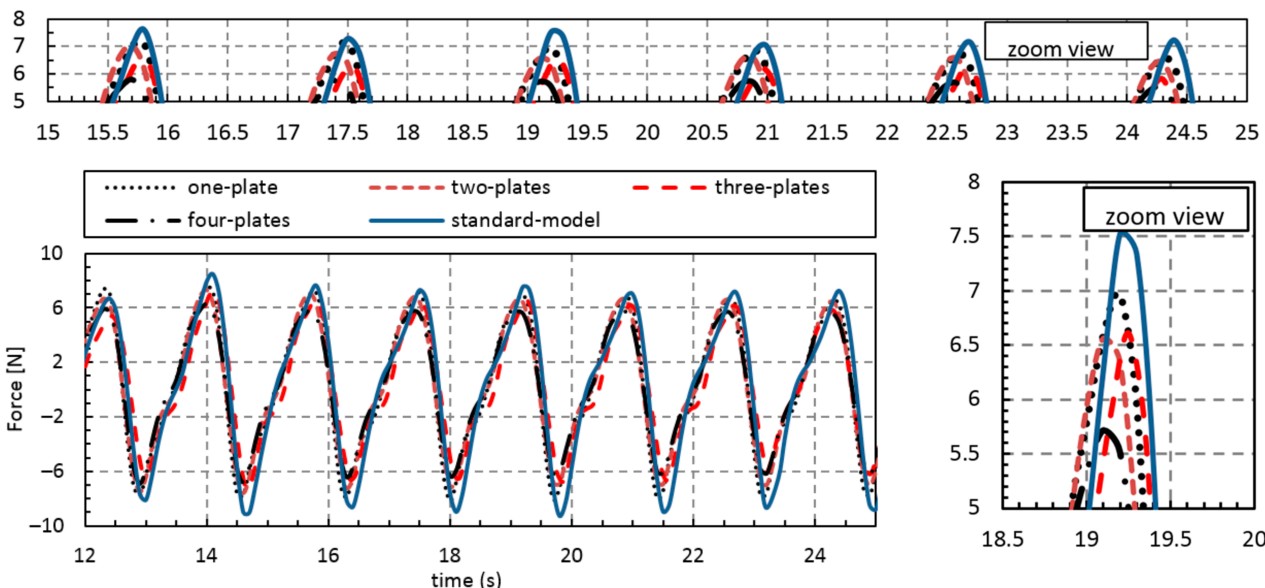

**Figure 22.** Comparing the force applied to the front wall of multi-plates/OWC for different numbers of installed plates, Kd = 1.26.

## 4. Conclusions

The present study's framework introduced a low-cost and applicable plan to increase efficiency further and decrease the wave load exerted on the converter structure. The plan is to divide the single chamber into multiple sub-chambers using vertical curtain plates. Although all these changes are evaluated only for one OWC as a sample, this plan could be expanded for other OWCs with different geometries. In this research, the open-source CFD code "InterFoam" from the OpenFOAM libraries performed a detailed hydrodynamic analysis of an OWC. The numerical solution approach is divided into two parts; first, similar to every numerical solution, the results are verified based on published experimental data. The quantitative derivation of numerical results is about e = 6.5% for efficiency on average.

Second, the curtain plates are added to the chambers' converter one by one, and the numerical solution is performed for each case. The results indicate that the converter's total efficiency increased at the particular wavelength for all cases (one, two, three, and four plates). For the wavelengths Kd = 0.5 to Kd = 0.7 and Kd = 1.8 to Kd = 2.5, the values for the efficiency deteriorated, but in the functional range of Kd = 0.7 to Kd = 1.8, the efficiency increased from e = 71% in the standard model to e = 88% for the most efficient case. Thus, the overall efficiency, which is important for a long-term period, increased significantly. These positive hydrodynamic effects are due to eliminating the undesired free surface sloshing effects and decreasing the phase difference between the maximum pressure and free surface velocity. Another question is to what extent adding these plates influenced the exerted loads on the structure. The results indicate that one of the main sections of the convertors' structure, called the front wall (lip), bore fewer loads when vertical plates are added to the chamber. Among these considered cases, four plates/OWC recorded the highest reduction rate, around 30%; this is because the loads are shared on the vertical plates. Overall, the results presented are based on the RANS method for a new and innovative multi-sub-chamber OWC with a shared orifice. Further investigations to evaluate other kinds of OWCs are still needed.

**Author Contributions:** Data collection by M.M. and M.Y.; data curing by M.M. and M.Y.; Conceptualization by M.M., A.M. and M.Y.; wrting the original draft by M.M. and M.Y.; review and editing by M.M., M.Y. and A.M.; administration and validation by A.M. All authors have read and agreed to the published version of the manuscript.

**Funding:** This research is funded by the project GINOP-2.2.1-18-2018-00015.

**Institutional Review Board Statement:** Not applicable.

**Informed Consent Statement:** Not applicable.

**Data Availability Statement:** The data can be obtained through the corresponding author.

**Acknowledgments:** The support from the project GINOP-2.2.1-18-2018-00015 is acknowledged.

**Conflicts of Interest:** The authors declare no conflict of interest.

## Nomenclature

**Latin Letters**

| | | | |
|---|---|---|---|
| a | Chamber lip height (m) | dx | Plate's thickness (m) |
| B | Chamber breadth (m) | wfs | Free surface velocity (m.s$^{-1}$) |
| C | OWC thickness (m) | Kd | Wave-number (-) |
| d | Water depth (m) | dB | Orifice width (m) |
| H | OWC height (m) | P* | Pseudo-dynamic pressure (Pa) |
| Z | Height of plates (m) | $P_c$ | Air pressure (Pa) |
| b | Sub-chamber width (m) | $P_{in}$ | Power of the incident wave (W) |
| k | Angular wave-number (m$^{-1}$) | $P_{out}$ | Power available at air outlet (W) |
| F | Force (N) | q | Air volume flow rate at turbine (m$^3$.s$^{-1}$) |
| g | Acceleration of gravity (m.s$^{-2}$) | $R_{dev}$ | Deviatoric stress tensor |
| h | Wave height (m) | $S_{f,i}$ | Face area vector |
| U | Velocity vector (m.s$^{-1}$) | E | error (%) |
| T | Wave period (s) | e | efficiency value (%) |
| X | Position vector (m) | EL | Free surface elevation |
| V | Turbulent viscosity | | |

**Greek Letters**

| | | | |
|---|---|---|---|
| $\lambda$ | Wave length (m) | $\varepsilon$ | Kinetic energy dissapation rate |
| $\nu$ | Fluid kinematic viscosity | $\omega$ | Angular frequency |
| $\tau_{ij}$ | Reynolds stress term | $\theta$ | Phase difference |
| $\kappa$ | Turbulent kinetic energy | $\eta$ | efficiency symbol |
| $\rho$ | Density (kg.m$^{-3}$) | $\xi$ | Wave steepness (-) |
| $\nu_t$ | Fluid turbulent kinematic viscosity | $C_\mu$ | Constant (in two-equations model) |
| $\sigma_k, \sigma_\varepsilon, C_{\varepsilon1}, C_{\varepsilon2}$ | Constant (in modeled dissipation rate transport equation) | | |

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
