# Peer review of "Efficiency Assessment of an Amended Oscillating Water Column Using OpenFOAM"

_sustainability, doi:10.3390/su13105633_

Round 1
Reviewer 1 Report
The authors addressed some of this reviewers’ comments, but not the most important:
- Figure 3. All plots show a blue dash-dot line. The legend calls it “theory”. Apparently, it is a theoretical wave amplitude. Please, define it in the main text, or which theory is used to calculate those values.
- Lines 350 – 357. The authors added the following sentences: “The effective mechanism whereby all the parameters could be evaluated is using a comprehensive symbol. For this reason, disparity symbol engaged to illustrate each parameters’ deviation for amended OWC rather than standard OWC in each case-studies framework, Figure 18. only for three–plate/OWC, particularly from kd=0.7 to kd=1.5, the pressure increases, but the pressure had a reverse trend in other cases. A notable inference is, apart from the number of plates, the efficiency increases from kd=0.7 to kd=1.8. That is to say, the hydrodynamic performance of the device can be improved for some specific wave frequencies, which are near the natural frequency.” The definition of disparity percentage is not yet clear. Please, give an equation with the definition.
Minor remark.
Very often the periods “.” are not followed by capital letters. Please, check these editorial typos before the submission.
Author Response
Dear Professor,
Thank you for the evaluation of the work precisely. We have revised accordingly and our revisions are highlighted for your kind consideration. As you advised, we have extended the recommendation section to include more details. We are grateful for your kind consideration and time.
Response to Referee:
General comment:
- Figure 3. All plots show a blue dash-dot line. The legend calls it “theory”. Apparently, it is a theoretical wave amplitude. Please, define it in the main text or which theory is used to calculate those values.
Authors' answer:
“ This paragraph is added to the revised manuscript, from line 202 to 209.”
The lines for the theory in the paper comes from the wave amplitude, and the free surface height from the bottom, lower bond=0.92-0.06=0.86 m, and upper bond=0.92+0.06=0.98 m, are specified to evaluate the accuracy of the numerical solution. Although each case's results are similar, to get the accuracy with the low computational cost, dx=0.05m could be a reasonable choice for the case in-hand.
Response to Referee:
General comment:
- The definition of disparity percentage is not yet clear. Please, give an equation with the definition.
Authors' answer:
As previously mentioned in the manuscript, the disparity percentage is only a symbolic parameter to represent the deviations in each case; this parameter could be explained by using these equations:
the disparity percentage calculated by substituting the three main parameters, p: pressure, wfs: free surface velocity, E: efficiency for standard, and amended OWC into the equation below:
(a) |
(b) |
(C) |
“ from line 370 to 379 in the revised manuscript.”
Response to Referee:
General comment:
Minor remark.
Very often the periods “.” are not followed by capital letters. Please, check these editorial typos before the submission.
Authors' answer:
Many linguistic and grammatical errors are corrected, some phrases are amended, and some are deleted. A double-checked is performed for the manuscript.
- I hope that the corrected manuscript satisfies and convinces the reviewers’ viewpoint

Reviewer 2 Report
The numerical model is not well explained and conducted. It may result in limited interest for the typical reader of Sustainability.

Author Response
Dear Professor,
Thank you for the evaluation of the work precisely. We have revised accordingly and our revisions are highlighted for your kind consideration. As you advised, we have extended the recommendation section to include more details. We are grateful for your kind consideration and time.
Response to Referee:
General comment:
As the reviewer collected and mentioned the points in the PDF manuscript, the answers are directly inscribed on that file.
Furthermore, it is noted that in the reviewer's response, there were many correcting parts highlighted in the revised manuscript version.
Many linguistic and grammatical errors are corrected, some phrases are re-organized, and some of them are deleted. A double-checked is performed for the manuscript.
- I hope that the corrected manuscript satisfies and convinces the reviewer’s viewpoint
We also did the native proofreading and double-checked all the equations. Further elaborations had been done and validations had been extended.

Round 2
Reviewer 1 Report
The authors addressed the technical remarks of this reviewer. Please check the following minor correction:
Lines 209, 282, 371: “Error! Reference source not found” appears in the text. Please, fix the reference.
Figure 8: please, make sure that all subplots are printed in the same page
Author Response
Dear Professor,
We are grateful to you for your positive comments and supportive feedback on the manuscript. Thank you very much for your constructive feedback and advice. We have carefully considered your valuable queries to improve the research further. In the following, we have replied to your queries and, our revisions are highlighted in the manuscript for your kind consideration.
We hope our revisions are satisfactory and can meet your expectation. We are ready to address your future comments if further revisions are required.
General comment:
- The authors addressed the technical remarks of this reviewer. Please check the following minor correction:
- Lines 209, 282, 371: “Error! Reference source not found” appears in the text. Please, fix the reference.
- Figure 8: please, make sure that all subplots are printed in the same page
Authors' answer:
The minor errors that you mentioned are corrected, and a double-checked is performed for the manuscript.
- I hope that the corrected manuscript satisfies and convinces the reviewers’ viewpoint
Reviewer 2 Report
Please find attached my comments.

Author Response
We are grateful to you for your positive comments and supportive feedback on the manuscript. Thank you very much for your constructive feedback and advice. We have carefully considered your valuable queries to improve the research further. In the following, we have replied to your queries and, our revisions are highlighted in the manuscript for your kind consideration. Enclosed, please kindly find the supplementary PDF.
We hope our revisions are satisfactory and can meet your expectation. We are ready to address your future comments if further revisions are required.
General comment:
As the reviewer collected and mentioned the points in the PDF manuscript, the answers are directly inscribed on the attached file enclosed.
Furthermore, it is noted that in the reviewer's response, there were many correcting parts highlighted in the revised manuscript version.
Many linguistic and grammatical errors are corrected, some phrases are re-organized, and some are deleted. A double-checked is performed for the manuscript.
- I hope that the corrected manuscript satisfies and convinces the reviewer’s viewpoint
We also did the native proofreading and double-checked all the equations. Further elaborations had been done, and validations had been extended.

Round 3
Reviewer 2 Report
The paper has been improved substantially about clarity of contents and aims. Still, the paper is ridden with grammar and syntax errors, mispellings, and the like. Should I receive again a paper with so many evident errors, I will suggest declining even if I still think it may be a very interesting paper if it is written with more care. Find attached a few minor suggestions.

Author Response
Dear Professor,
We have carefully considered all your comments. The Native Proofreading had been also been applied. We reconsider all the comments to ensure all the revisions had been effectively done.
Sincerely yours,
Dr. Amir Mosavi
This manuscript is a resubmission of an earlier submission. The following is a list of the peer review reports and author responses from that submission.
Round 1
Reviewer 1 Report
Sustainability
Efficiency assessment of an amended surrogate model for conventional oscillating water column using OpenFOAM
by Mobin Masoomi, Mahdi Yousefifard and Amir Mosavi
Reviewer’s report
General comments
The manuscript illustrates a numerical investigation carried out to evaluate the influence of vertical plates inside an OWC chamber on the performance of the device.
Although the topic is of some interest to the scientific community, the paper has several flaws and cannot be published in its current form.
Authors analyzed just one OWC configuration and do not justify the selected geometrical parameters. Therefore, the results cannot be generalized.
Authors carried out numerical simulation, so it is not understandable why a scale model is made.
Moreover, the performed analyses are affected by a gross error. Equation 10 is valid only for deep water conditions.
Finally, the whole article is full of oversights and grammatical errors that make it difficult to read. Therefore, it must be reviewed by a native English speaker.
Minor points:
- The abstract is not clear and should be revised.
- OWCs, like other WEC devices, are not widely used. There are very few installations around the world right now and probably none are operational.
- The introduction lacks the reference to recent relevant paper on OWC device (i.e U-OWC). Furthermore, a more effective bibliography research on the wave force on OWC device should be carried on.
- Some symbols are adopted for different parameters.
- All references should be carefully checked and the bibliography format should be compiled according to journal standards.
- Please check all the adopted symbol (i.e. “K” at line 129)
- Table 1 What is “OWC height (H)? Please indicate such a parameter in Figure 1
- The paper has several parts which are redundant. Please move the literature referment to the introduction.
Reviewer 2 Report
This article investigates the use of OWC chamber partition on efficiency and, to a limited extent, on the structural load. The article proposes an interesting analysis, which makes it eligible for publication. I suggest a major revision for improving the quality of the paper. In particular, I strongly suggest to revise the text against typos, grammatical errors and formatting issues. The readability of the current article is difficult in some parts. In addition, the following remarks must be addressed:
Line 31: “the take-off system” is usually called “the power take-off”. I suggest to conform to the terminology used in the field.
Line 46: amend the citation format in “Takahashi et al. Takahashi, Nakada”.
Figure 3: a curve labeled “theory” is included. What does it represent? It does not look useful, eventually I suggest to remove it.
Line 316-317: the authors write “The lowest and highest efficiencies are recorded in free surface piston mode and sloshing mode, respectively”, but I think that they wanted to write the vice-versa.
Line 373-374: I suggest to remove the sentence “A complete illustration of how the pressure changes for 373 different kd is shown in Figure 17”. The reader is going to see that plot later.
Figure 17: please define “disparity percentage”.
Reviewer 3 Report
The aim of this manuscript is the numerical investigation on the performance of an Oscillating Water Column (OWC) Wave Energy Converter. RANS equations and VOF method were used in OpenFOAM to solve the problem. The topic addressed is of interest in the field of renewable and sustainable energy, with some practical implications. However, I do not think that the paper could be published in the present form. In my opinion, the numerical model is not well explained and conducted, and the paper must be thoroughly revised before a deeper review. In the present version, it may result in limited interest for the typical reader of Sustainability.